# Bootstrapping Top-down Information for Self-modulating Slot Attention

**Dongwon Kim**[1]  **Seoyeon Kim**[1]  **Suha Kwak**[1,2]
Dept. of CSE, POSTECH[1]  Graduate School of AI, POSTECH[2]
{kdwon, syeonkim07, suha.kwak}@postech.ac.kr

## Abstract

Object-centric learning (OCL) aims to learn representations of individual objects within visual scenes without manual supervision, facilitating efficient and effective visual reasoning. Traditional OCL methods primarily employ bottom-up approaches that aggregate homogeneous visual features to represent objects. However, in complex visual environments, these methods often fall short due to the heterogeneous nature of visual features within an object. To address this, we propose a novel OCL framework incorporating a *top-down pathway*. This pathway first bootstraps the semantics of individual objects and then modulates the model to prioritize features relevant to these semantics. By dynamically modulating the model based on its own output, our *top-down pathway* enhances the representational quality of objects. Our framework achieves state-of-the-art performance across multiple synthetic and real-world object-discovery benchmarks.

## 1 Introduction

Object-centric learning (OCL) is the task of learning representations of individual objects from visual scenes without manual labels. The task draws inspiration from the human perception which naturally decomposes a scene into individual entities for comprehending and interacting with the real world visual environment. Object-centric representations provides improved generalization and robustness [9], and have been proven to be useful for diverse downstream tasks such as visual reasoning [44], simulation [45], and multi-modal learning [24]. In this context, OCL which learns such representations without labeled data has gained increasing attention.

A successful line of OCL builds upon slot attention [30]. This method decomposes an image into a set of representations, called slots, that iteratively compete with each other to aggregate image features. Reconstructing the original image from the slots, they are encouraged to capture entities constituting the scene. This simple yet effective method has been further advanced by novel encoder or decoder architectures [33, 19, 46, 42], optimization technique [18, 6], and new query initialization strategies [18, 24].

It is worth noting that all these methods are fundamentally considered bottom-up models, as they rely on aggregating visual features without incorporating high-level semantic information from the beginning. This bottom-up approach assumes that visual features within an object are homogeneous and can be clustered in the feature space, which only holds for simplistic objects that can be identified using low-level cues such as color [20]. In complex real-world scenarios where visual entities of the same semantics exhibit diverse appearances, this homogeneity often breaks down, leading to suboptimal object representations [22, 47]. Thus, we take an approach different from the previous line of research: *introducing top-down information into slot attention*, such as object categories and semantic attributes.

38th Conference on Neural Information Processing Systems (NeurIPS 2024).

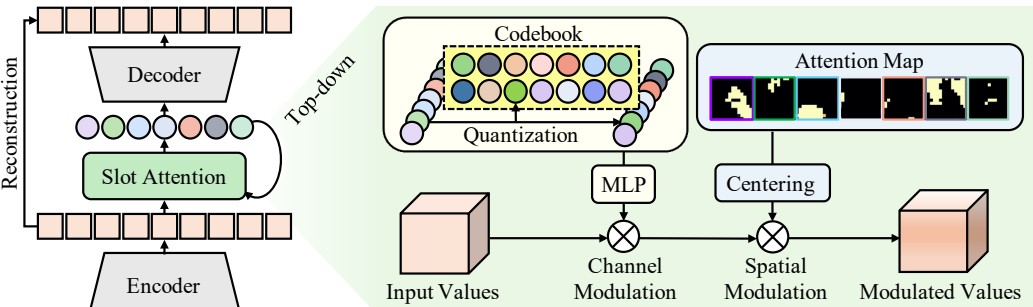

Figure 1: The overall pipeline of our framework. A *top-down pathway* is introduced into slot attention to utilize top-down information. The pathway consists of two parts: bootstrapping top-down knowledge and exploiting them. Firstly, semantic information is bootstrapped from slot attention outputs by mapping slots to discrete codes from a learned codebook through vector quantization. Secondly, slot attention is modulated using these codes and its attention maps, transforming it into a self-modulating module. Inner activations are modulated across channels with codes and across space with centered attention maps. Slot attention is then repeated with these modulated activations, yielding more representative slots.

Incorporating top-down information enables slot attention to specialize in discerning objects within specific semantic categories. For instance, identifying vehicles in a complex urban environment can be challenging due to the diverse and cluttered nature of the scene. Top-down information can guide the model to prioritize vehicle-specific features, such as wheels and windows. This inhibits the contributions of irrelevant features when computing slots, and enhances the aggregation of visual features of individual vehicles into slots. Nevertheless, devising such a top-down approach is not straightforward since OCL assumes an unsupervised setting without any labeled data, making it hard to identify and exploit the high-level semantics typically obtained from annotated datasets.

We propose a novel framework that incorporates a *top-down pathway* into slot attention to provide and exploit top-down semantic information; Fig. 1 illustrates of our framework. The pathway consists of two parts: bootstrapping semantics and exploiting them for better representations. Firstly, top-down semantic information is bootstrapped from the output of slot attention itself, by mapping continuous slots to discrete codes selected from a finite learned codebook. Such an approach allows the codebook to learn prevalent semantics in the dataset, with each code representing a specific semantic concept. Thus, semantic information can be bootstrapped without any object-level annotations and used to provide top-down semantic information. Secondly, slot attention is modulated using bootstrapped top-down cues obtained from the first phase, which we call self-modulation. In this phase, the *top-down pathway* dynamically guides the slot attention by re-scaling its inner activations based on the top-down information. This self-modulation process enables the model to focus on feature sub-spaces where object homogeneity is more consistent, thereby improving its performance in diverse and realistic settings.

Our contributions are threefold:

- We introduce a method to bootstrap top-down semantic information from the output of slot attention, without requiring any object-level annotations. This allows the extraction of high-level semantic cues from an unsupervised learning process.
- We propose a self-modulation scheme that dynamically guides the slot attention's inner activations to enhance object representation, successfully incorporating the top-down semantic cues extracted.
- By integrating the proposed *top-down pathway* into slot attention, we demonstrate that the performance of object discovery is largely improved on various OCL benchmarks.

## 2 Related Work

**Object-centric learning** OCL aims to learn representations of individual objects within an image. The 'object-centric' dimension is orthogonal to the conventional representation learning which learns representations independent of the composition of the image. The structured nature of

object-centric representations offers improved generalization [9], making it valuable for various applications, including visual reasoning [44], dynamics simulation [45], and multi-modal learning [23, 24]. A foundational method in this field is slot attention [30], which introduced a simple yet effective framework that employs a competitive attention mechanism between slots. Following slot attention, many recent works have proposed improvements by introducing novel encoder or decoder formulations [33, 19, 46, 35], optimization techniques [18, 6], additional slot refinement modules [37, 25, 3], and expansions to video modality [27, 11, 36]. These methods are primarily bottom-up models, while our approach proposes to bootstrap and incorporate top-down information.

**Incorporating top-down information** The human visual system perceives scenes by leveraging both top-down and bottom-up visual information [4, 8]. Top-down information represents task-driven contextual cues, such as high-level semantics and prior knowledge about the scene. In contrast, bottom-up information is derived directly from the sensory input. Inspired by this dual-processing mechanism of the human visual system, several studies [34, 1, 48] have attempted to model this approach within deep learning, achieving significant improvements across various tasks. Our work follows in a similar direction, specifically focusing on introducing top-down information into the representative OCL method, slot attention. By incorporating top-down semantic and spatial information, we aim to enhance the performance of slot attention in diverse visual environments, addressing the limitations of previous bottom-up methods

**Discrete representation learning** Discrete representations within neural networks are considered effective for modeling discrete modalities [50, 38] and tackling generation tasks [13, 28]. Particularly, the pioneering work, VQ-VAE [38], introduced a method for learning discrete latent representations through vector quantization. This model uses a discrete codebook, where the encoder maps input data to discrete codes using nearest-neighbor lookup and the decoder reconstructs the input from the codes. Another notable approach is the Gumbel-softmax trick [17, 32], which provides a differentiable approximation of sampling from a categorical distribution. Recent advancements have aimed to incorporate a more sophisticated formulation of codebooks [49] or improve codebook utilization [16] to better handle discrete representations. Recent work by Wallingford et al. [40] is related to our research in terms of using vector quantization for segmentation task. However, our method differs in that the quantized codes are used to modulate bottom-up slot attention, while in Wallingford et al. [40], the codes are used solely for segmentation labeling.

## 3 Method

We propose an OCL framework that incorporates top-down semantic information, such as object categories and semantic attributes, into slot attention through a *top-down pathway*. Fig. 1 illustrates the overall pipeline of our framework. Firstly, slot attention is applied to visual features extracted from an image encoder to output slots (Sec. 3.1). Then, a *top-down pathway* leverages the slots to identify semantics in the input image and modulate slot attention. The pathway consists of two parts: bootstrapping top-down semantic information from a learned codebook and attention maps (Sec. 3.2) and modulating the inner activations of slot attention with this semantic information (Sec. 3.3). During the self-modulation stage, slot attention is repeated with the modulated activations, resulting in more representative slots.

### 3.1 Slot Attention

Slot attention is a recurrent bottom-up module that aggregates image features into slots through an iterative attention process, where each of the resulting slots represent an entity in the input image. Within our framework, these slots are used to bootstrap top-down information in the later stage (Sec. 3.2).

The module takes in the initial slots, $S^0 \in \mathbb{R}^{K \times D}$, and visual features extracted from an image encoder, $\mathbf{x} \in \mathbb{R}^{N \times D_{\text{feat}}}$. The initial slots are obtained by sampling $K$ vectors from a learnable Gaussian distribution using a reparameterization trick. The slots $S = [s_1, s_2, \dots, s_K] \in \mathbb{R}^{K \times D}$, where each represents an individual object in the image, are computed by iteratively updating the initial slots $T$ times as

$$S := S^T, \text{where} \ \ S^{t+1} = \texttt{slot\_attn}(\mathbf{x}, S^t). \tag{1}$$

In each iteration, the slots attend to the visual features, refining their representations through a series of attention-based updates. Let $q(\cdot)$, $k(\cdot)$, and $v(\cdot)$ represent linear projections from dimension $d$ to

$d_h$. Then, the attention map $\boldsymbol{A} \in \mathbb{R}^{K \times N}$ is computed as

$$\boldsymbol{A}_{i,j} = \frac{e^{P_{i,j}}}{\sum_{i=1}^{K} e^{P_{i,j}}}, \quad \text{where } P = \frac{q\left(\boldsymbol{S}^{t-1}\right)k\left(\mathbf{x}\right)^{\top}}{\sqrt{d_h}}. \tag{2}$$

Unlike the original attention introduced in transformer [39] which normalizes across the keys, the attention in slot attention is normalized across the slots. Such a distinct normalization scheme makes the slots compete with each other to aggregate the visual features, encouraging each slot to represent a distinct object in the scene. Then, the computed attention map $\boldsymbol{A}$ is normalized across the rows into $\tilde{\boldsymbol{A}}$ and used to update the slots as

$$\boldsymbol{S}^t = f_{\texttt{update}}(\boldsymbol{U}, \boldsymbol{S}^{t-1}), \quad \text{where } \boldsymbol{U} = \tilde{\boldsymbol{A}}v\left(\mathbf{x}\right), \tag{3}$$

such that $\boldsymbol{U}$ is the weighted sum of the visual features $v(\mathbf{x})$. The function $f_{\texttt{update}}$ processes $\boldsymbol{U}$ with consecutive GRU [7] and Multi-Layer Perceptron (MLP) and residually sums it to the previous slots, $\boldsymbol{S}^{t-1}$, to produce the updated slots, $\boldsymbol{S}^t$. For further details, refer to Locatello et al. [30].

### 3.2 Bootstrapping Top-down Information

Our idea to bootstrap top-down information without annotations is based on our observation that the slots $\boldsymbol{S}$, which are outputs of the bottom-up attention module, contain rough semantic information about objects. We leverage this coarse information to bootstrap both the semantic and spatial top-down information about the objects coarsely represented by the slots. Top-down semantic information pertains to the specific semantic categories or attributes of the objects (*what*), while top-down spatial information indicates the locations or regions within the image where these objects are located (*where*). Incorporating such knowledge can guide slot attention to focus on the features most relevant to the objects expected to appear, enabling it to accurately capture objects that are obscured or have high intra-object variance, such as people with different hairstyles or clothing.

Firstly, we extract the "*what*" information from the slots $\boldsymbol{S}$ using Vector Quantization (VQ), which maps each slot to one of the semantic concepts learned throughout training. Specifically, each slot $\boldsymbol{S}_k$ is mapped to the nearest code in a finite codebook, $\mathbb{C} = [\boldsymbol{c}_1, \boldsymbol{c}_2, \ldots, \boldsymbol{c}_E] \in \mathbb{R}^{E \times D}$ with size $E$. The mapped code $\boldsymbol{c}_k^* \in \mathbb{R}^D$ is considered a top-down semantic cue for the slot $\boldsymbol{s}_k$. Formally, this quantization process can be written as

$$\boldsymbol{c}_k^* = \underset{\boldsymbol{c} \in \mathbb{C}}{\arg\min} \|\boldsymbol{s}_k - \boldsymbol{c}\|_2^2. \tag{4}$$

Since the $\arg\min$ operation is non-differentiable, we use the straight-through estimator [2] for backpropagation. During training, the codebook learns to store distinct semantic patterns recurring within the dataset by quantizing continuous slot embeddings into a limited number of discrete embeddings. Thereby, each code can act as automatically discovered top-down semantic information.

Secondly, we obtain the "*where*" information from the attention maps of the last layer of slot attention. For each slot $\boldsymbol{s}_k$, the $k$-th row vector of the attention map $\boldsymbol{A}$, denoted as $\boldsymbol{a}_k \in \mathbb{R}^N$, is used to aggregate visual features and update $\boldsymbol{s}_k$. This attention map provides useful spatial prior information about where each extracted top-down semantic information is located in the image.

### 3.3 Self-modulating Slot Attention

In the original slot attention (Sec. 3.1), the slot updates are driven purely by visual features extracted from the input without incorporating higher-level semantic information that can provide additional context. To address this limitation, we introduce self-modulating slot attention, which modulates the computation of the slot updates based on the top-down information obtained in the bootstrapping stage (Sec. 3.2). This bootstrapped top-down information is used to dynamically amplify or inhibit specific channel dimensions or regions of the value-projected visual features, while keeping the model parameters unchanged. Formally, self-modulating slot attention can be represented by conditioning slot attention with the vector quantized slots $\boldsymbol{c}_k^*$ and their corresponding slot-wise attention map $\boldsymbol{a}_k$:

$$\hat{\boldsymbol{S}} := \hat{\boldsymbol{S}}^T, \text{where } \hat{\boldsymbol{S}}^{t+1} = \texttt{slot\_attn}\left(\mathbf{x}, \hat{\boldsymbol{S}}^t; [\boldsymbol{c}_i^*]_{i=1}^K, [\boldsymbol{a}_i]_{i=1}^K\right), \tag{5}$$

**Algorithm 1** Self-modulating Slot Attention. The module inputs visual features extracted from an encoder, initial slots sampled from a learned Gaussian distribution, and $K$ vector quantized slots and their corresponding $K$ slot-wise attention maps output from the original slot attention. The number of iterations, $T$, is set to three.

---

**Input:** Visual features $\mathbf{x}$, initial slots $\hat{\boldsymbol{S}}^0$, vector quantized slots $[\boldsymbol{c}_i^*]_{i=1}^K$, and slot-wise attention maps $[\boldsymbol{a}_i]_{i=1}^K$.
**Output:** Slots after $T$-iteration of self-modulating slot attention $\hat{\boldsymbol{S}}^T$.
1: **for** $k = 1$ to $K$ in parallel **do**
2:   $\boldsymbol{m}_k^c = \texttt{MLP}(\boldsymbol{c}_k^*)$               // Compute channel-wise modulation vector
3:   $\boldsymbol{m}_k^s = 1 + (\boldsymbol{a}_k - \overline{\boldsymbol{a}_k})$         // Compute spatial-wise modulation vector
4:   $\boldsymbol{M}_k = \boldsymbol{m}_k^s \otimes \boldsymbol{m}_k^c$              // Compute modulation map
5: **for** $t = 1$ to $T$ **do**
6:   $\tilde{\boldsymbol{A}} = \texttt{softmax}\big(q(\hat{\boldsymbol{S}}^{t-1})k(\mathbf{x})^\top / \sqrt{D_h}\big)$
7:   $\hat{\boldsymbol{S}}^t = f_{\texttt{update}}([\boldsymbol{u}_1, \boldsymbol{u}_2, \ldots, \boldsymbol{u}_K], \hat{\boldsymbol{S}}^{t-1})$, where $\boldsymbol{u}_k = \tilde{\boldsymbol{A}}_k(\boldsymbol{M}_k \odot v(\mathbf{x}))$    // Modulated slot update
8: **return** $\hat{\boldsymbol{S}}^T$

---

where $\hat{\boldsymbol{S}}$ represents the slots from the self-modulating slot attention, different from the slots of the original slot attention denoted by $\boldsymbol{S}$. Note that the original slot attention (Sec. 3.1) and self-modulating slot attention share parameter weights and initial slots, such that $\boldsymbol{S}^0 = \hat{\boldsymbol{S}}^0$.

Specifically, we modulate slot attention with a modulation map $\boldsymbol{M}_k \in \mathbb{R}^{N \times D}$ computed from $\boldsymbol{c}_k^*$ and $\boldsymbol{a}_k$. Each element of the modulation map represents the relevance score between the corresponding visual feature element and the top-down information of the expected object. This modulation map can be used to guide the update of each slot $\hat{\boldsymbol{S}}_k$ (Eq. 3), by prioritizing specific value elements with the high relevance scores. In the self-modulating slot attention, computation of the slot update $\boldsymbol{U}$ is replaced with:

$$\boldsymbol{U} = [\boldsymbol{u}_1, \boldsymbol{u}_2, \ldots, \boldsymbol{u}_K] \in \mathbb{R}^{K \times D}, \ \ \boldsymbol{u}_k = \tilde{A}_k(\boldsymbol{M}_k \odot v(\mathbf{x})), \tag{6}$$

where $\odot$ represents Hadamard product. Such re-scaling of the value features with the modulation map ensures that the specific channel dimension or regions contribute more to the update of each slot, based on the semantics or locations encoded in bootstrapped top-down information.

The modulation map $\boldsymbol{M}_k$ is computed by taking the outer product between channel-wise and spatial-wise modulation vectors, which are predicted using $\boldsymbol{c}_k^*$ and $\boldsymbol{a}_k$, respectively:

$$\boldsymbol{M}_k = \boldsymbol{m}_k^s \otimes \boldsymbol{m}_k^c \in \mathbb{R}^{N \times D}. \tag{7}$$

For predicting channel-wise modulation vector $\boldsymbol{m}_k^c$, quantized slot $\boldsymbol{c}_k^*$ is used, which tells us "*what*" the object appearing in the image is. The channel-wise scaling is designed to enforce the model to focus on certain feature subspaces closely correlated to the semantic concept identified. Specifically, channel-wise modulation vector $\boldsymbol{m}_k^c$ can be obtained by feeding quantized slot $\boldsymbol{c}_k^*$ to the MLP, which is represented as:

$$\boldsymbol{m}_k^c = \texttt{MLP}(\boldsymbol{c}_k^*) \in \mathbb{R}^D. \tag{8}$$

The spatial-wise modulation vector $\boldsymbol{m}_k^s$ can be obtained by further processing the attention map of each slot, which contains the top-down information on the location of the semantic concept:

$$\boldsymbol{m}_k^s = 1 + (\boldsymbol{a}_k - \overline{\boldsymbol{a}_k}) \in \mathbb{R}^N, \tag{9}$$

where $\overline{\boldsymbol{a}_k}$ is for the average of the attention score of $\boldsymbol{a}_k$. Using an attention map as is for modulation will make all values down-scaled, while some regions likely to contain the object should be highlighted for effective incorporation of the spatial top-down information. Thus, we use the attention map shifted to have a mean value of 1 for the spatial-wise modulation map.

## 3.4 Training

Slot attention is trained within an autoencoding framework, using a decoder that reconstructs visual features output by the image encoder [35, 33] or the original image [30] from the slots. In this paper, we choose the visual feature reconstruction as our training objective since it is known to provide more robust training signals for real-world datasets [33]. We also employ a vector quantization objective

for the codebook $\mathbb{C}$ only, which thereby learns to minimize the mean-squared error between the slot and the sampled codes. The reconstruction objective $\mathcal{L}_{\text{recon}}$ and vector quantization objective $\mathcal{L}_{\text{VQ}}$ are given by

$$\mathcal{L}_{\text{recon}} = \|\text{Dec}(\hat{\boldsymbol{S}}; \mathbf{x}) - \mathbf{x}\|_2^2, \quad \mathcal{L}_{\text{VQ}} = \|\text{sg}(\boldsymbol{S}) - \boldsymbol{C}^*\|_2^2, \tag{10}$$

where $\text{sg}(\cdot)$ represents stop gradient operation and $\boldsymbol{C}^* = [\boldsymbol{c}_1^*, \boldsymbol{c}_2^*, \ldots, \boldsymbol{c}_K^*] \in \mathbb{R}^{K \times D}$. For the decoder, we utilized the autoregressive slot decoder [35, 33]. The reconstruction objective ensures that the learned slot representations capture essential information about the objects in the scene, while the vector quantization objective encourages the codebook to capture recurring semantic concepts in the dataset.

# 4 Experiments

## 4.1 Experimental Settings

**Datasets** To verify the proposed method in diverse settings, including synthetic and authentic datasets, we considered four object-centric learning benchmarks: MOVI-C [15], MOVI-E [15], PASCAL VOC 2012 [14], and MS COCO 2017 [29]. MOVI-C and MOVI-E are synthetic datasets, adopted for validating our method in relatively simple visual environments. MOVI-C contains 87,633 images for training and 6,000 images for evaluation, while MOVI-E contains 87,741 and 6,000, respectively. To evaluate the proposed model in real-world settings, we leverage the VOC and COCO datasets. Following DINOSAUR [33], we use the `trainaug` variants, containsing 10,582 training images, for VOC dataset. For the evaluation, we use the validation split containing 1,449 images. The COCO dataset consists of 118,287 training images and 5,000 images for evaluation. While the VOC dataset includes some images with a single object, images of the COCO dataset always contain 2 or more objects, making it the most challenging. MOVI datasets are licensed under apache license 2.0 and COCO is licensed under CC-BY-4.0.

**Metrics** We evaluate our method with three metrics: foreground adjusted random index (FG-ARI), mean best overlap (mBO), and mean intersection over union (mIoU). The FG-ARI is the ARI metric computed for foreground regions only (objects), which measures the similarity between different clustering results. The mBO and mIoU are both IoU-based metrics, computed for all regions including the background. The mBO computes the average IoU between ground truth and prediction pairs, obtained by assigning each prediction to the ground truth mask with the largest overlap. The mIoU is computed as the average IoU between ground truth and prediction pairs obtained from Hungarian matching. For COCO and VOC, $\text{mBO}^i$ and $\text{mBO}^c$ indicate the mBO metric computed using semantic segmentation and instance segmentation ground truth. We use the instance segmentation ground truth for other metrics. Following previous work [33], the internal attention maps of the autoregressive decoder are used as the mask prediction results of the slots.

**Implementation details** To assess the effectiveness of the proposed top-down pathway, our model is implemented based on DINOSAUR [33], a representative slot-based OCL method. For the encoder and decoder, we use a DINO [5] pretrained ViT-B/16 [10] and an autoregressive transformer decoder [35, 33], respectively. The model is trained using an Adam optimizer [26] with an initial learning rate of 0.0004, while the encoder parameters are not trained. The number of slots $K$ is set to 11, 24, 7, and 6 for MOVI-C, MOVI-E, COCO, and VOC, respectively. The codebook size $E$ is set to 128 for synthetic datasets (MOVI-C and MOVI-E) and 512 for authentic datasets (COCO and VOC). The model is trained for 250K iterations on VOC and for 500K iterations on the others. For the ablation study and analysis, models are trained for 200K iterations on COCO, as this was enough to reveal overall trends given limited computational resources. Full training of the model takes 26 hours using a single NVIDIA RTX 3090 GPU.

**Codebook size $E$ selection** The performance of the proposed *top-down pathway* depends on codebook size $E$ (Sec. 4.4), necessitating a principled selection method. We determine $E$ automatically by monitoring the perplexity of code usage distribution during training, requiring only the training set without validation data. Perplexity—the exponent of entropy—indicates how uniformly the codes are being used. While perplexity typically increases with codebook size, it plateaus when $E$ exceeds the number of distinct semantic patterns in the data, as some codes become unused [16, 49, 28]. To find the optimal size, we start with $E = 64$ and double it until the perplexity plateaus after 250K iterations. For example, on COCO, the perplexity when the codebook size is 256, 512, and 1024 are

Table 1: Comparison with DINOSAUR [33] on synthetic datasets: MOVI-C [15] and MOVI-E [15]. We include both the reported and reproduced performance of DINOSAUR for fair comparison.

| Method | MOVI-C | | | MOVI-E | | |
|---|---|---|---|---|---|---|
| | FG-ARI | mBO$^i$ | mIoU | FG-ARI | mBO$^i$ | mIoU |
| DINOSAUR [33] | 55.7 | 42.4 | - | - | - | - |
| DINOSAUR reprod | 54.7±4.1 | 41.9±1.8 | 41.0±2.1 | 53.8±2.1 | 34.5±1.7 | 33.6±1.9 |
| Ours | **58.9**±5.1 | **46.8**±2.4 | **45.9**±2.5 | **59.7**±3.1 | **39.3**±1.8 | **38.3**±1.9 |

Table 2: Comparison with DINOSAUR [33] on real-world datasets: COCO [29] and VOC [14]. We include both the reported and reproduced performance of DINOSAUR for fair comparison.

| Method | COCO | | | | VOC | | | |
|---|---|---|---|---|---|---|---|---|
| | FG-ARI | mBO$^i$ | mBO$^c$ | mIoU | FG-ARI | mBO$^i$ | mBO$^c$ | mIoU |
| DINOSAUR [33] | 34.1±1.0 | 31.6±0.7 | 39.7±0.9 | - | 24.8±2.2 | 44.0±1.9 | 51.2±1.9 | - |
| DINOSAUR reprod [33] | 34.1±0.9 | 31.4±0.5 | 39.5±0.1 | 29.4±0.6 | **27.0**±3.2 | 41.2±3.4 | 48.2±3.8 | 39.0±3.6 |
| Ours | **37.4**±0.0 | **33.0**±0.3 | **40.3**±0.2 | **31.2**±0.3 | 26.7±4.7 | **43.9**±2.6 | **51.0**±2.5 | **42.0**±2.8 |

176.9, 253.9, and 242.8, respectively, where 512 was chosen as the final size. This procedure enables efficient hyperparameter selection using only training data, eliminating the need for validation set tuning. Following this approach, we set $E = 128$ for synthetic datasets (MOVI-C and MOVI-E) and $E = 512$ for real-world datasets (COCO and VOC).

## 4.2   Quantitative Analysis

DINOSAUR [33] is the first successful OCL method that scales slot attention to real-world datasets by introducing the use of a self-supervised image encoder [5], an autoregressive decoder, and a feature reconstruction objective. Notably, DINOSAUR uses the vanilla slot attention mechanism without modifications, making it a perfect baseline for validating the effectiveness of our proposed *top-down pathway*. Thus, we adopt DINOSAUR as a baseline and compare its performance with and without our proposed method. For a fair comparison, we report both the reported and reproduced performance of DINOSAUR.

Tab. 1 demonstrates that incorporating the proposed *top-down pathway* into DINOSAUR largely improves performance in every metric. Specifically, our method improves FG-ARI by 5.9 on MOVI-E, which is the most challenging synthetic dataset. In Tab. 2, performances on authentic datasets, COCO and VOC, are reported. Our method largely surpasses the reproduced baseline on most metrics. The only metric for which our method does not show improvement is the FG-ARI on VOC. We hypothesize that this is because VOC images frequently contain single objects only, and FG-ARI is computed solely with foreground pixels so that the performance is less affected by the top-down information.

Tab. 3 presents a comparison between our method and recent state-of-the-art methods. It is notable that our proposed method achieves competitive performance even to recent methods using advanced diffusion-based decoders [19, 46]. Moreover, our approach focuses on incorporating top-down information into slot attention, which is orthogonal to the line of work advancing decoders to provide better training signals to slot attention.

## 4.3   Qualitative Results

**Codebook visualization**  To validate whether the proposed codebook learns meaningful semantic concepts, we present visualizations of the codebook in Fig. 2. The index of the code and the mask prediction obtained from the slots modulated by the code are presented together, revealing the semantic entity each code represents. Visualization demonstrates that the codebook successfully discovers and stores distinct semantic concepts without using any annotations. Moreover, the codes are mapped to objects with various appearances and layouts, which demonstrates that the codes learn high-level semantic information and not low-level structural or positional information.

**Prediction visualization**  In Fig. 3, we visualize the original image, mask predictions, and slot attention maps $A$ before and after self-modulation. Results demonstrate that the modulation dynamically

Table 3: Comparison with state of the arts [33] on COCO [29], VOC [14], MOVI-C [15], and MOVI-E [15]. Performances of Slot Attention on MOVI-C and -E are reproduced by [33] and that of SLATE by [19]. Those on COCO and VOC are from [46].

| Method | COCO | | VOC | | MOVI-C | | MOVI-E | |
|---|---|---|---|---|---|---|---|---|
| | FG-ARI | mBO$^i$ | FG-ARI | mBO$^i$ | FG-ARI | mBO$^i$ | FG-ARI | mBO$^i$ |
| Slot Attention [30] | 21.4 | 17.2 | 12.3 | 24.6 | 43.8±0.3 | 26.2±1.0 | 45.0±1.7 | 24.0±1.2 |
| SLATE [35] | 32.5 | 29.1 | 15.6 | 35.9 | 49.54±1.4 | 39.4±0.8 | 46.06±3.3 | 30.2±1.7 |
| DINOSAUR [33] | 34.1±1.0 | 31.6±0.7 | 24.8±2.2 | 44.0±1.9 | 55.7 | 42.4 | - | - |
| Rotating Features [31] | - | - | - | 40.7±0.1 | - | - | - | - |
| SlotDiffusion [46] | 37.2 | 31.0 | 17.8 | **50.4** | - | - | **60.0** | 30.2 |
| LSD [19] | 35.0 | 30.4 | - | - | 52.0 | 45.6 | 52.2 | 39.0 |
| Ours | **37.4**±0.0 | **33.3**±0.3 | **26.7**±4.7 | 43.9±2.6 | **58.9**±5.1 | **46.8**±2.4 | 59.7±3.1 | **39.3**±1.8 |

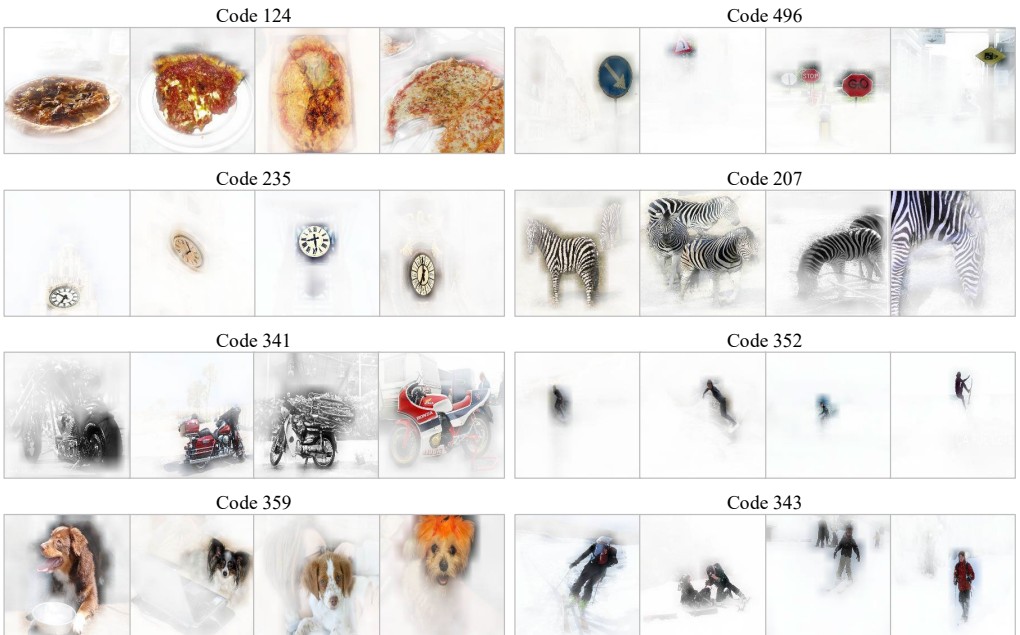

Code 124                                    Code 496

Code 235                                    Code 207

Code 341                                    Code 352

Code 359                                    Code 343

Figure 2: Visualization of the codebook $\mathbb{C}$ on COCO [29]. The results show that the codebook learns to capture recurring semantic concepts in the dataset, such as 'pizza' (code 124), 'sign' (code 496), 'clock' (code 235), 'zebra' (code 207), 'motorcycle' (code 341), 'surfer' (code 352), 'dog' (code 359), and 'skier' (code 343).

refines the attention maps, depending on how well they have captured the scene. For example, when the attention map is well-structured but coarse, the modulation process refines the boundaries of the attention map without changing the overall layout (first row). However, if the attention maps fail to delineate objects, the modulation process recomposes the attention maps to differentiate objects (second to fourth row). By providing top-down semantic and spatial information via self-modulation, the attention maps are enhanced to capture the object within complex real-world environments.

### 4.4 In-depth Analysis

**Effect of codebook size** In our framework, vector quantization maps slots to distinct top-down semantic information stored in the codebook, as shown in Fig. 2. In Tab. 4, we report the performances across different codebook sizes, $E$. The results show that a codebook size too large ($E = 1024$) or small ($E \leq 256$) leads to performance degradation. When the codebook size is too small, the codes cannot sufficiently learn distinct semantic information, which degrades the quality of the bootstrapped top-down semantic information. On the other hand, a codebook size too large may cause the codes to capture irrelevant details such as appearance variance or positional information rather than meaningful semantic concepts. However, we determine the optimal codebook size automatically using the perplexity of codebook usage during training (Sec. 4.1), eliminating the need for extensive hyperparameter tuning using validation split and benchmark metrics.

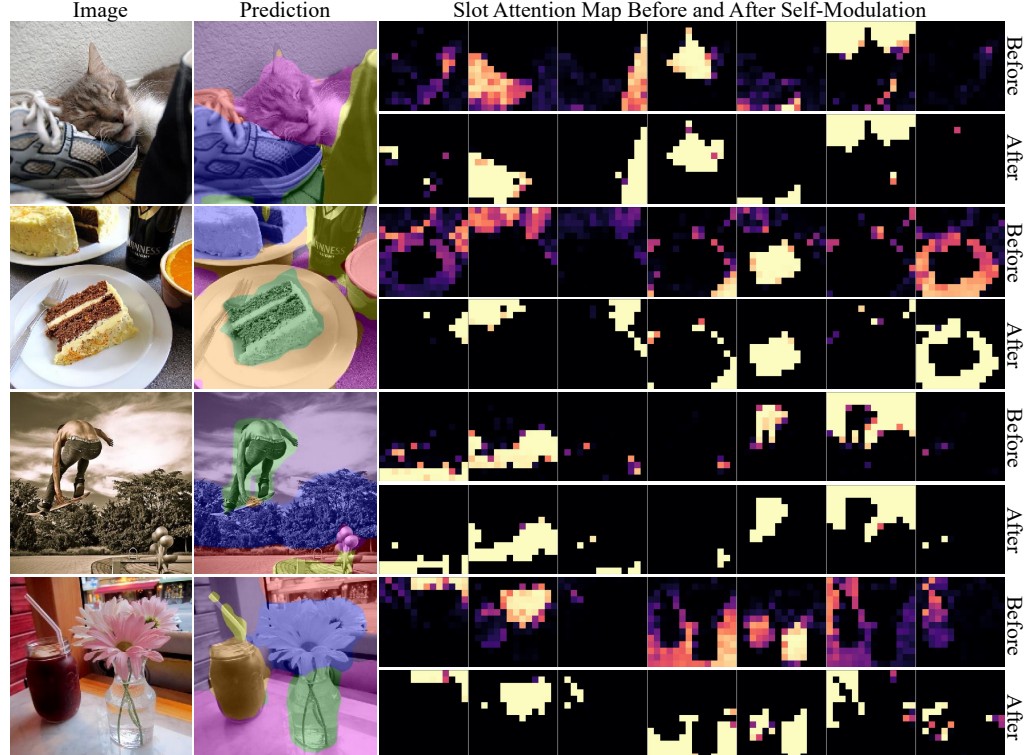

Figure 3: Visualization of the input image, predicted object mask, and attention maps of slot attention before and after self-modulation on COCO [29]. Lighter the color, higher the attention score.

Table 4: Comparison between different codebook sizes on COCO [29].

|  | 128 | 256 | 512 | 1024 |
|---|---|---|---|---|
| FG-ARI | 33.1 | 32.6 | **37.3** | 31.4 |
| mBO | 30.2 | 30.5 | **32.7** | 29.8 |

Table 5: Comparison with DINOSAUR [33] using six iterations in slot attention on COCO [29].

|  | FG-ARI | mBO |
|---|---|---|
| DINOSAUR 6-iter | 28.5 | 28.2 |
| Ours | **37.3** | **32.7** |

**Impact of increased iterations** Since our method requires repeating slot attention with self-modulation, we investigate whether our performance improvement simply comes from the increased iterations of slot attention. In Tab. 5, we compare the results of our model with the DINOSAUR baseline model using six iterations of slot attention, which is twice as many iterations as the default setting. Notably, both our model and the DINOSAUR 6-iteration model leverage the same number of iterations. The results show that the DINOSAUR model with six iterations actually performs worse compared to the default 3 iterations. This indicates that merely increasing the number of iterations does not guarantee improvement. Our method's superior performance is thus attributed to the self-modulation mechanism rather than the increased number of iterations.

**Computation overhead of *top-down pathway*** While our model requires one more forward pass for slot attention, the additional computation cost is negligible. In DINOSAUR, slot attention accounts for only 0.64% of the total FLOPs, compared to 71.26% for the encoder and 28.10% for the decoder. Consequently, our model requires 47.62 GFLOPs versus 47.32 GFLOPs of DINOSAUR—a mere increase below 1%. In practice, processing the entire COCO 2017 val split on a single NVIDIA RTX 3090 GPU takes 71.4 seconds for our model compared to 70.5 seconds for DINOSAUR, demonstrating minimal impact on inference time.

**Codes representing broader semantics** While most codes in our codebook consistently represent single object categories, we observed interesting cases where codes capture broader concepts, as shown in Fig. 4. Some codes are trained to represent supercategories - for instance, grouping different animal species (code 468) or various human parts into shared codes (code 328). This suggests the codebook can flexibly adapt to different levels of semantic abstraction when beneficial. We also discovered an edge case where certain codes (e.g., code 223) specialize in capturing top-left patches of images. This behavior appears to be influenced by the autoregressive decoding process, which

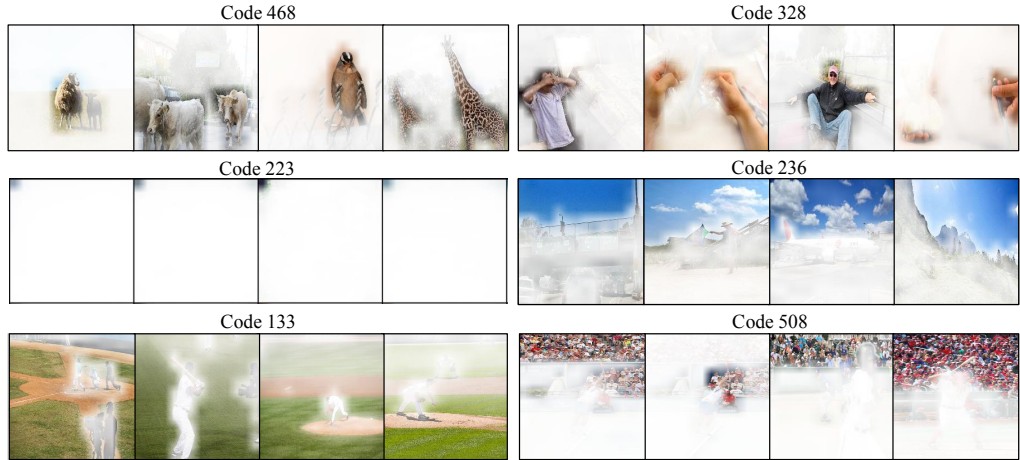

Figure 4: Visualization of the codebook $\mathbb{C}$ on COCO [29]. The results show that the codebook learns to capture broader semantics other than single object categories, such as supercategory (code 468, 328), top-left patch (code 223), and background (code 236, 133, 508).

must reconstruct the top-left patch first without surrounding context. However, these specialized positional codes are rare (1-2 out of 512 codes) and have minimal impact on overall performance. Additionally, we found that certain codes specialize in capturing background elements common in natural scenes. For example, code 236 represents sky regions, code 133 captures sports fields, and code 508 represents crowd scenes.

**Ablation study** In Tab. 6, we present an ablation study of our method on COCO for channel-wise modulation, vector quantization, spatial-wise modulation, and attention map shifting. In the first row, the result with no modules is presented, which is equivalent to the baseline DINOSAUR model. The last row is the performance of using all four modules, equivalent to our proposed method. The second and third rows are for the ablation of vector quantization and channel-wise modulation, demonstrating that incorporating top-down semantic information significantly improves performance. In the fourth and fifth rows, the ablation of attention map shifting and spatial-wise modulation is presented. The results show that both operations are critical for the effective exploitation of top-down spatial information.

Table 6: Ablation studies on COCO dataset for each module consisting of the proposed top-down pathway: channel-wise modulation ($m^c$), vector quantization (VQ), spatial-wise modulation ($m^s$), and shifting attention map (shift).

| $m^c$ | VQ | $m^s$ | shift | FG-ARI | mBO |
|:---:|:---:|:---:|:---:|:---:|:---:|
| DINOSAUR | | | | 34.8 | 30.5 |
| ✓ | | ✓ | ✓ | 36.3 | 32.3 |
| | | ✓ | ✓ | 35.1 | 32.5 |
| ✓ | ✓ | ✓ | | 35.2 | 31.7 |
| ✓ | ✓ | | | 36.0 | 31.9 |
| ✓ | ✓ | ✓ | ✓ | **37.3** | **32.7** |

## 5 Conclusion

In this paper, we introduced an OCL framework that incorporates top-down information into the slot attention mechanism through a *top-down pathway*. In this pathway, the output of the slot attention is used to bootstrap high-level semantic knowledge and rough localization cues for existing objects. Using the bootstrapped top-down knowledge, slot attention is modulated to focus on features most relevant to the objects in the scene. Consequently, by incorporating the proposed *top-down pathway* into slot attention, we achieved state-of-the-art performance on various OCL benchmarks, including challenging synthetic and authentic datasets.

**Limitation** The proposed *top-down pathway* has a limitation in that its overall performance relies on the quality of the codebook learned during training. As shown in Tab. 4, an incorrect choice of codebook size can result in the codes failing to learn distinct semantic concepts or capturing irrelevant details. While we mitigate this limitation through perplexity-based automatic codebook size tuning, the more principled codebook design that can eliminate the need for a pre-defined hyperparameter, such as dynamically expanding codebook during learning [40], will be promising future research direction.

## Acknowledgments and Disclosure of Funding

This work was supported by IITP grants funded by the Korea government (MSIT) (RS-2019-II191906 Artificial Intelligence Graduate School Program (POSTECH); RS-2024-00457882 AI Research Hub Project; RS-2024-00509258 Global AI Frontier Lab).

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

# Appendices

## A    Additional Qualitative Results

Fig. A6 demonstrates the additional visualizations of original image, mask predictions, and slot attention maps $A$ before and after self-modulation on the COCO dataset.

## B    Details of Autoregressive Decoder

The proposed *top-down pathway* is implemented based on the DINOSAUR baseline [33], using an autoregressive slot decoder. Singh et al. [35] first proposed to use such an autoregressive decoding scheme for slot attention training. Autoregressive decoder is known to provide better training signal leading to improved performance, compared to the MLP-based broadcast decoder [43] used by the slot attention originally. Autoregressive decoder is the simple variant of the transformer decoder [39], which takes input visual feature $\mathbf{x}$ and slots $S$. The decoder consists of multiple decoding blocks. Let multi-head attention be denoted as $\mathtt{MHA}(Q; K; V)$, where $Q$, $K$, and $V$ is for query, key, and value, respectively. Then, the decoding block can be represented as:

$$\mathtt{DecBlock}(\mathbf{x}; S) = \mathtt{FFN}(\mathtt{MHA}(\tilde{\mathbf{x}}; S; S)), \tag{11}$$

$$\tilde{\mathbf{x}} = \mathtt{MHA}_{<}(\mathbf{x}_{[\mathtt{BOS}]}; \mathbf{x}_{[\mathtt{BOS}]}; \mathbf{x}_{[\mathtt{BOS}]}), \tag{12}$$

where $\mathtt{FFN}$ denotes a feedforward layer with MLP and residual connection, $\mathtt{MHA}_{<}(\cdot)$ represents multi-head self-attention with causal masking, and $\mathbf{x}_{[\mathtt{BOS}]}$ represents the visual feature sequence with a learnable $[\mathtt{BOS}]$ token appended at the start of the sequence. By using multiple decoding blocks, we can compute the autoregressive reconstruction of the visual feature $\mathbf{x}$, which is consequently used for the computing reconstruction objective. Following DINOSAUR, we use an autoregressive decoder with four decoding blocks. The number of heads for multi-head attention is set to 8.

## C    Additional Experiments

***Top-down pathway* without DINO and autoregressive decoder** We have mainly built a *top-down pathway* with the DINO [5] pretrained weight and autoregressive decoder (framework proposed in DINOSAUR [33]) since it is the best working object-centric learning framework in a real-world setting.

To see if these settings are essential for the *top-down pathway*, we have implemented our self-modulation technique with the original slot attention setting [30], which includes training encoders from scratch and using an image reconstruction objective with spatial broadcast decoder [43]. Tab. A7 summarizes the results of the CLEVR6 dataset. We observe a significant improvement in mBO, showing that our self-modulation technique is applicable to slot attention and provides complementary benefits. Although FG-ARI decreased, mBO is considered more robust when evaluating model performance [25, 12, 21, 33]. We have also included qualitative results in Fig. A5, which demonstrates that self-modulation markedly improves segmentation. These results indicate our method's effectiveness with different encoder configurations and training objectives.

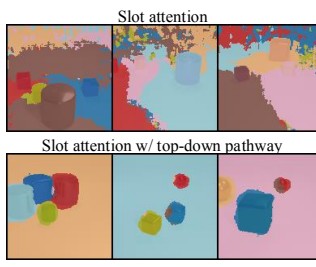

Figure A5: Visualization of the predicted object mask on CLEVR6 [20].

**Comparison to MaskCut [41]** While object-centric learning (OCL) and unsupervised instance segmentation share the goal of discovering objects without supervision, their ultimate objectives differ. OCL aims to learn object-wise representations that support downstream tasks requiring compositionality and systematic generalization, whereas unsupervised instance segmentation focuses primarily on obtaining accurate object masks. Nevertheless, we can directly compare methods from both tasks on their object discovery capabilities. We evaluate our method against MaskCut, the pseudo-mask generation algorithm underlying CutLER [41], on the COCO dataset. As shown in Tab. A8, our method significantly outperforms MaskCut across all metrics, demonstrating its effectiveness for object discovery even when compared to specialized unsupervised segmentation approaches.

Table A7: Comparison with slot attention [30] on CLEVR6

| | FG-ARI | mBO |
|---|---|---|
| Slot attn (reprod.) [30] | 98.7 | 21.2 |
| Slot attn (reprod.) + top-down pathway | 88.7 | 61.9 |

Table A8: Comparison with MaskCut [41] on COCO

| | FG-ARI | $mBO_i$ | mIoU |
|---|---|---|---|
| MaskCut [41] | 31.5 | 28.9 | 26.7 |
| Ours | $37.4 \pm 0.0$ | $33.0 \pm 0.3$ | $31.2 \pm 0.3$ |

Table A9: Comparison with SPOT [21] on COCO [29], VOC [14], MOVI-C [15], and MOVI-E [15].

| Method | COCO | | VOC | | MOVI-C | | MOVI-E | |
|---|---|---|---|---|---|---|---|---|
| | FG-ARI | $mBO^i$ | FG-ARI | $mBO^i$ | FG-ARI | $mBO^i$ | FG-ARI | $mBO^i$ |
| SPOT [21] | 36.6 ±0.3 | **34.7** ±0.1 | 19.4 ±0.7 | **48.1** ±0.4 | 52.1 ±3.3 | **47.0** ±1.2 | 56.4 ±4.1 | **39.9** ±1.1 |
| Ours | **37.4**±0.0 | 33.3±0.3 | **26.7**±4.7 | 43.9±2.6 | **58.9**±5.1 | 46.8±2.4 | **59.7**±3.1 | 39.3±1.8 |

**Comparison with SPOT [21]** In Tab. A9, we present the comparison with SPOT [21], a recent state-of-the-art object-centric learning method. SPOT proposed various ideas that can improve the quality of object-centric representation, such as patch order permutation within autoregressive decoder and self-distillation. We want to emphasize that these ideas are all orthogonal to the proposed *top-down pathway* and can be used together for further improvement, which we will leave as future work.

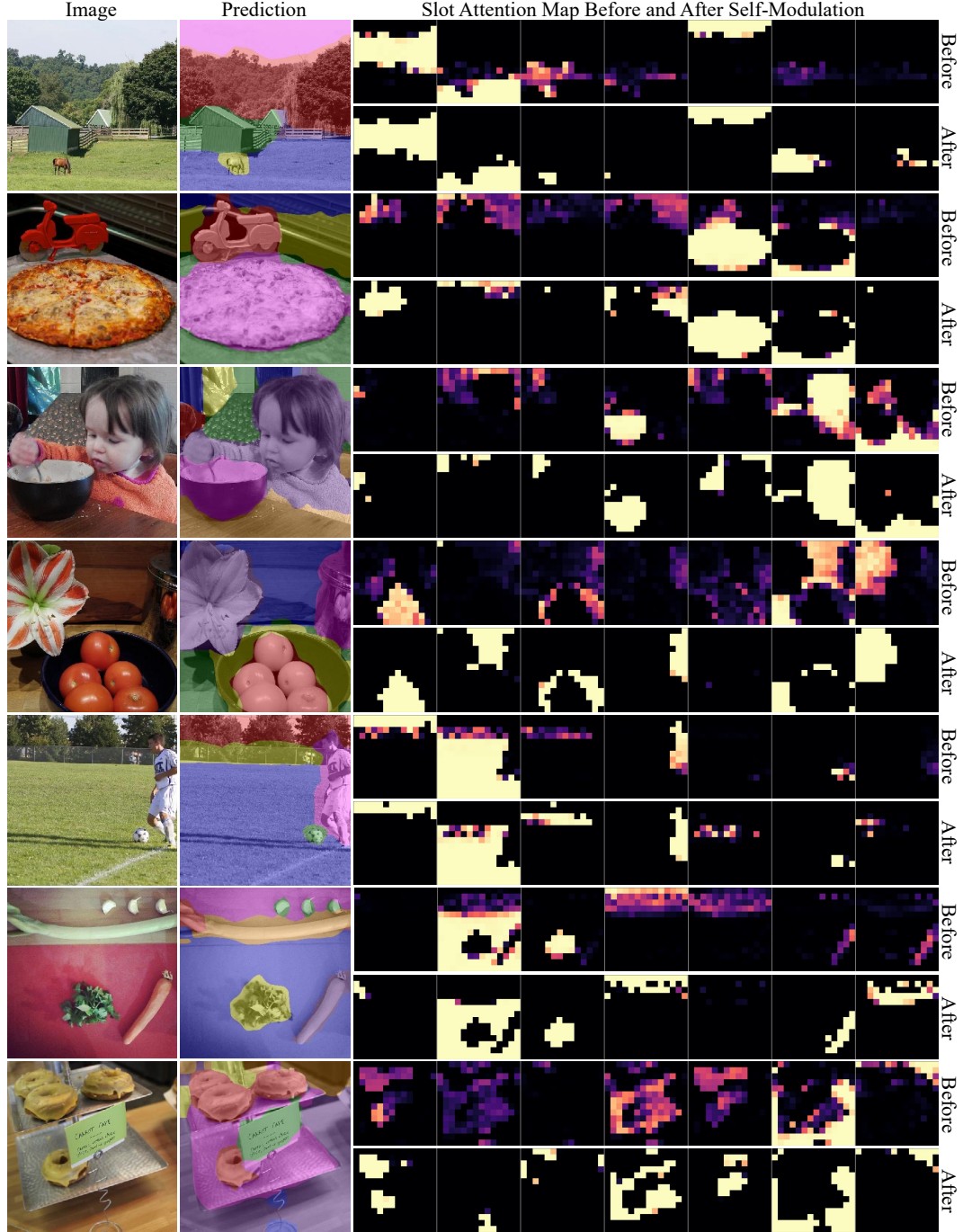

Figure A6: Visualization of the input image, predicted object mask, and attention maps of slot attention before and after self-modulation on COCO [29]. Lighter the color, higher the attention score.

