# OpenReview forum: "Bootstrapping Top-down Information for Self-modulating Slot Attention"
_NeurIPS.cc/2024/Conference — NeurIPS 2024 poster_

### Official Review · Reviewer_PJTV · 2024-07-07

**Soundness:** 3
**Presentation:** 3
**Contribution:** 3
**Rating:** 5
**Confidence:** 4

**Summary:**

This paper proposes a method that improves the performance of object-centric learning by modulating Slot Attention with semantic and location information obtained based on the output slots and attention maps of Slot Attention. For a given output slot, the semantic information is chosen as the vector closest to this slot in a codebook that is learned from all output slots of the entire dataset via vector quantization, and the location information is chosen as the shifted attention map with a mean value of 1. The proposed method is implemented based on DINASOUR and is compared with DINASOUR and other methods on two synthetic and two real datasets. The proposed method outperforms the compared method in most cases when the size of codebook is chosen appropriately.

**Strengths:**

1. Improving the performance of object-centric learning with top-down information is an important and interesting research direction.
2. The proposed method outperforms the compared methods when the size of codebook is chosen appropriately.

**Weaknesses:**

1. My main concern about the paper is that the performance of the proposed method is very sensitive to the size of codebook. If the size of codebook is not chosen appropriately, the performance can be even worse than the original DINASOUR method. Moreover, how to choose the best size of codebook is not described clearly.
2. From my understanding, the core of the proposed method is a self-modulation module (along with a codebook) that is compatible with all the object-centric learning methods using the slot attention mechanism. However, the proposed method is only implemented based on DINASOUR. The quality of the paper could be significantly improved if the proposed method is implemented based on at least another object-centric learning method that is developed based on slot attention.
3. The proposed self-modulation includes both semantic and spatial modulations. However, the codebook used in the semantic modulation also contains spatial information. This design can be more elegant, and the codebook can be much more useful if it only contains semantic information. In this way, whether two objects belong to the same category can be determined automatically.
4. Some data in the experimental part are inconsistent. For example, the mBO^i of the proposed method on COCO is 33.0 in Table 2, 33.3 in Table 3, and 32.7 in Table 4.
5.  The authors attribute both semantic modulation and spatial modulation as top-down modulations. I agree that the semantic modulation is a top-down modulation because the codebook contains knowledge learned from all images (not just the inferred image). However, I don’t think that the spatial modulation can be considered top-down because it just uses attention maps computed in a bottom-up manner (only based on the content of one image).

**Questions:**

1. How is the best size of codebook chosen?
2. Is it possible to learn a codebook that is independent of the locations of objects in the image?

**Limitations:**

The authors have addressed the limitation in the conclusion part, and there is no potential negative societal impact.

---

> ### Author Rebuttal · Authors · 2024-08-07
>
> ## Sensitivity to codebook size
> We acknowledge the performance dependence on the codebook size, as noted in our limitations section.
> However, we would like to emphasize that the optimal codebook size was determined automatically using the training set only (without the validation set). To choose the codebook size, we monitor the perplexity of the code usage distribution during training. Perplexity—the exponent of entropy—indicates how uniformly the codes are being used and typically increases as the codebook becomes larger. However, when the codebook size exceeds the number of semantic patterns in the data, some codes become neglected, causing low code utilization [D, 45, 26]. Hence, the ideal codebook size leads to the maximum perplexity.
>
> To find the optimal codebook size, we start with a size of 64 and monitor the perplexity of the codebook over training. After sufficient iteration (e.g., 250K), the codebook size is doubled repeatedly until the perplexity reaches a plateau. For example, on COCO, the perplexity when the codebook size is 256, 512, and 1024 are 176.9, 253.9, and 242.8, respectively, where 512 was chosen as the final size. Thereby, an adequate codebook size could be efficiently determined without using the validation set.
>
> ## Experiment on models other than DINOSAUR
> We appreciate your suggestion and have implemented our self-modulation technique with the original Slot Attention setting [28], which trains the encoder from scratch and uses an image reconstruction objective. The following table summarizes the results on the CLEVR6 dataset:
> |  | FG-ARI | mBO |
> |----------|----------|----------|
> | Original slot attention | 98.7 | 21.2 |
> | Original slot attention + SelfMod | 88.7 | 61.9 |
>
> We observe a significant improvement in mBO, showing that our self-modulation technique is applicable to slot attention and provides complementary benefits. Although FG-ARI decreased, mBO is considered more robust and should be prioritized when evaluating model performance, as detailed in our response to reviewer 3ckb ("Necessity & effect of VQ: (3) Difference in FG-ARI and mBO in VQ Ablation"). We have also included qualitative results in Figure R3 of the attached PDF, which demonstrates that self-modulation markedly improves segmentation. These results indicate our method's effectiveness with different encoder configurations and training objectives, suggesting it does not heavily rely on pretrained features. We will add the results and discussion in the revision.
> ## Does the codebook also contain positional information?
> Our codebook primarily encodes semantic content, largely independent of the object position (Figure 2 of the  main paper). We hypothesize that this is because semantic information dominates slot representations, with positional information occupying a small subspace. This is supported by $K$-means clustering of DINOSAUR slot representations, which automatically groups semantic categories using simple L2 distance (Figure R6 of the attached PDF). While our approach empirically extracts mainly semantic information, explicitly disentangling positional and semantic information remains an interesting future direction, with work like [3] offering promising leads.
> ## Inconsistent numbers between tables
> We apologize for the confusion. Our mBO$_i$ on COCO of 33.3 in Table 3 is a typo, and the correct number is 33.0. We will revise the paper to include the correct measurement. Despite this error, our method still outperforms the LSD [18] by 2.6%p, showing the effectiveness of our approach.
>
> Additionally, as mentioned in line 234 on page 6, the in-depth analysis and ablation studies (Tables 4, 5, and 6) were conducted with models trained for 200K iterations, rather than the full 500K iterations, due to computational constraints. This explains the differences between Tables 1-3 and Tables 4-6. We found that the models mostly converged at 200K iterations and thus was enough to reflect the performance trends of models trained for the full 500K iterations. To improve clarity, we will update the captions of Tables 4-6. We appreciate your attention to these details.
> ## Can spatial modulation be called “top-down”?
> We appreciate the reviewer’s comment on our top-down approach. We respectfully clarify our reasons for calling spatial modulation a top-down approach.
>
> A top-down approach is defined by the use of prior knowledge (contextual or task-relevant) to guide visual processing, while a bottom-up approach relies solely on sensory data. As the reviewer noted, our semantic modulation using vector quantization aligns with this definition, leveraging contextual knowledge learned from the dataset.
>
> Although the spatial information is derived from the same image, the spatial modulation provides rough object saliency maps to the subsequent slot attention, which is crucial task-relevant information for object discovery. It is task-relevant because it explicitly guides the model to discover objects within the provided saliency map, narrowing the search space. Such spatial cues are recognized as a form of top-down knowledge in human vision research [A, B, C], guiding attention to regions likely containing objects of interest.
>
> In conclusion, while our semantic and spatial modulations differ in their information sources, both embody top-down processing principles by providing crucial task-relevant guidance to the visual processing pipeline.
>
> ---
> [A] SUN: Top-down saliency using natural statistics, Visual cognition, 2009
>
> [B] Components of visual orienting, Attention and performance X: Control of language processes, 1984
>
> [C] Oculomotor strategies for the direction of gaze tested with a real-world activity, Vision research, 2003
>
> [D] Straightening Out the Straight-Through Estimator: Overcoming Optimization Challenges in Vector Quantized Networks, ICML, 2023

---

> > ### Comment · Reviewer_PJTV · 2024-08-10
> >
> > Thanks for the detailed response. Since the codebook size is determined solely based on the training set in an unsupervised way and applying the proposed self-modulation to Slot Attention also improves the performance on the CLEVR6 dataset, my concerns about technical flaws disappear and the rating is increased.

---

> > > ### Author Response · Authors · 2024-08-13
> > >
> > > We sincerely appreciate your thoughtful reconsideration of our work. We are pleased that our clarifications regarding codebook size determination and the CLEVR6 experiment have addressed your concerns. We will improve the paper in future revision based on your feedback. Thank you for your valuable feedback and updated assessment.

---

### Official Review · Reviewer_qEia · 2024-07-11

**Soundness:** 3
**Presentation:** 3
**Contribution:** 3
**Rating:** 6
**Confidence:** 4

**Summary:**

Slot Attention is a popular component for object-centric learning methods. In this
paper, the authors propose an extension of Slot Attention named "top-down pathway".
After the last iteration of Slot Attention, the slots are mapped to a discrete, learnable
codebook. Jointly with the final attention map, this information is used in a second
iteration of Slot Attention to modulate the QKV-like attention mechanism. Previous
models are consistently outperformed on established benchmarks. Moreover, a more
detailed analysis of the learned codebook shows that distinct semantic concepts are
represented by different codebook entries.

**Strengths:**

- The proposed extension of Slot Attention is clearly explained. The original Slot
  Attention is described at an adequate level of detail, which helps to make the paper
  accessible.
- The model consistently outperforms DINOSAUR, from which all design decisions are
  derived. This demonstrates the effectiveness of the novel mechanism.
- The analysis of the learned codebook is insightful and confirms the motivation of the
  proposed method.

**Weaknesses:**

I am not fully convinced by the term "top-down pathway". Only the recurrent Slot
  Attention module has been extended, the rest of the model is still bottom-up. From
  the initial description, I would have expected top-down modulation reaching to the
  encoder.

  Did you consider a variant of the model that continues with the slot embeddings from
  the first iteration instead of starting from scratch? If this performs similarly well,
  it might indiciate that the quantization due to the codebook is driving the improvement.

**Questions:**

- A recent approach to object-centric learning which is not based on Slot Attention is
  CutLER (Wang et al. 2023). How does the proposed method compare to CutLER?

**Limitations:**

Limitations are very briefly discussed. An additional point could be the dependence on
the pretrained feature encoder, which might not work well in all scenarios.

---

> ### Author Rebuttal · Authors · 2024-08-07
>
> ## Clarification on term “top-down pathway” / modulating encoder
> We appreciate the reviewer's comment on our use of the terminology, "top-down pathway." We respectfully explain that our terminology is appropriate for the following reasons:
> In our context, "top-down" refers to using higher-level task-relevant information to guide lower-level processing. Our method achieves this by: a) Extracting semantic information via vector quantization to enforce the focusing on certain features. b) Using spatial information from the previous slot attention to guide object localization.
> Both of the above mechanisms bootstrap top-down information beyond pure bottom-up processing, without requiring additional supervision. While our top-down pathway is confined to the slot attention, it still introduces a meaningful top-down influence on the features by modulating values in slot attention.
> We agree that extending the top-down modulation to the encoder is an interesting direction for future research. However, our current approach strikes a balance between effectiveness and efficiency, as modulating the encoder would significantly increase computational cost (the encoder accounts for over 71% of total compute). Our method achieves meaningful improvement without this additional overhead. For a detailed discussion of our method's efficiency, please refer to our response to Reviewer 3ckb (“Computation cost analysis”).
>
> ## Value modulation vs. slot initialization
> We appreciate the insightful suggestion. We did consider such a variant that initializes the slots with the quantized representations from the previous slot attention. However, this approach did not perform well in our experiments.
> Given that slot attention consists of 3 iterations, initializing the second slot attention with quantized representations from the first is equivalent to inserting VQ in the middle of 6 iterations. To evaluate this, we insert VQ into DINOSAUR's slot attention with 6 iterations, after the third iteration. Results on COCO with 200K training schedule are presented below, together with the results presented in the Table 5 of the main paper:
> |  | FG-ARI | mBO$_i$ |
> |----------|----------|----------|
> | DINOSAUR 6-iter  | 28.5 | 28.2 |
> | Ours  | 37.3 | 32.7 |
> | DINOSAUR 6-iter + VQ | 17.4 | 12.9 |
>
> Introducing VQ to DINOSAUR with 6 iterations leads to a performance decrease of 11.1%p and 15.3%p in FG-ARI and mBO, respectively (row 1 vs. 3 of the table). The severe performance degradation is likely due to the disruption in slot attention's convergence process. As shown in [6], slot representations naturally converge over its recurrent iterations. Discretizing in mid-process introduces a sudden shift in the slot representation, which will harm the overall optimization and convergence. Our method avoids this issue by modulating inner value activations, which can modulate the recurrent slot update process without directly changing the slot representation. By doing so, we can exploit the top-down information without hindering convergence of the slot attention.
>
> ## Comparison to CutLER
> We appreciate the suggestion to compare with CutLER [A], a recent approach to unsupervised object segmentation. While CutLER's full pipeline includes self-training of Mask R-CNN, we focus on comparing our method to CutLER's MaskCut algorithm for pseudo-mask generation, as this is the most direct comparison to our approach. Our method’s predictions can also be used as pseudo labels for training Mask R-CNN. We reproduced MaskCut using the official CutLER repository and evaluated it on the COCO dataset:
> |  | FG-ARI | mBO$_i$ | mIoU |
> |----------|----------|----------|----------|
> | MaskCut (CutLER )| 31.5 | 28.9 | 26.7 |
> | Ours | 37.4 $\pm$ 0.0 | 33.0 $\pm$ 0.3 | 31.2 $\pm$ 0.3 |
>
> Our method significantly outperforms MaskCut across all metrics, demonstrating its effectiveness for object discovery. We will add these results and discussion in the revision.
>
> ## Dependence on the pretrained feature encoder
> We appreciate the insightful suggestion. Regarding the inquiry, we conducted an experiment with the settings of the original slot attention [28], which trains the encoder from scratch (response to reviewer Yith, PJTV: “Experiment on models other than DINOSAUR”). Interestingly we observe substantial performance gain even in this setting, demonstrating that our method's improvements does not solely arise from using a pretrained feature encoder. For more detailed results and discussion, please refer to the corresponding response. We will discuss this experiment in the future revision as well.
>
>
> ---
> [A] Cut and Learn for Unsupervised Object Detection and Instance Segmentation, CVPR, 2023

---

> > ### Comment · Reviewer_qEia · 2024-08-12
> >
> > Thank you very much for the detailed response.
> >
> > - I am still not entirely convinced about the tmerin "top-down pathway", however I think
> >   this not a major issue that speaks against accepting the paper.
> > - The additional comparisons to MaskCut and other methods in the responses to other
> >   reviewers are very helpful and confirm the consistent improvements of the proposed
> >   model.
> > - I agree with the other reviewers on the concerns regarding the codebook size and the
> >   unclear role of the vector quantization. (1) While the codebook size can be chosen
> >   without supervision, it seems that the required multiple training runs substantially
> >   increase the computational cost. (2) The impact of the codebook on the performance
> >   seems to be small and heavily decreases the performance when used alone, as shown by
> >   the additional ablation study. While I still think the analysis of the codebook is
> >   interesting, I am not sure to which extent it is necessary for the performance
> >   improvement.
> >
> > Overall, I still think the paper should be accepted, since the proposed method is
> > interesting and consistently outperforms previous methods. Due to the remaining concerns
> > I keep my rating for now. But I am looking forward to read the other reviewers comments
> > on these points and happy to discuss further.

---

> > > ### Author Response · Authors · 2024-08-13
> > >
> > > We sincerely appreciate your detailed response and thoughtful consideration of our rebuttal. We will address your feedback as follows:
> > >
> > > - We will elaborate on the term "top-down pathway" in the paper to clarify its usage and context.
> > > - We're pleased you found the additional comparisons to MaskCut helpful. We will include this in the future revision.
> > > - Regarding codebook concerns: a) While determining optimal codebook size requires multiple training runs, perplexity is measured at half the full training schedule (250K), reducing computational burden. b) On the necessity of vector quantization, we kindly refer you to our response to reviewer 3ckb, specifically points (1), (3), and (4) under **"Necessity & effect of VQ"**, which provide further insights into VQ's role in our method's performance.
> > >
> > > We truly appreciate your valuable feedback and will incorporate your suggestions to improve our paper's final version. Thank you for your support!

---

### Official Review · Reviewer_Yith · 2024-07-12

**Soundness:** 3
**Presentation:** 3
**Contribution:** 3
**Rating:** 7
**Confidence:** 3

**Summary:**

This paper proposes a modification to Slot Attention incorporating top-down information into the algorithm. After an iteration of Slot Attention, the slots are quantized into a learned codebook. The quantized slots and attention maps are then used in another iteration of Slot Attention, refining the representations. The algorithm is evaluated in the DINOSAUR setting on the MOVi, Pascal VOC, and COCO datasets, showing improved segmentation quality over vanilla DINOSAUR. Visualizations of the codebook show that meaningful semantic concepts are learned and several ablations are performed.

**Strengths:**

The proposed method is well-motivated and seems to show an improvement over previous methods. The use of top-down semantic information from a learned codebook is novel, from my understanding. Overall, the paper is well-written and the authors provide a good analysis of their method including ablations of the different design choices.

**Weaknesses:**

1. The experiments are only performed in the DINOSAUR setting with pretrained, frozen ViT features as the input and reconstruction target. It would be very informative to also include experiments with the original image as the input and reconstruction target since that is also a common use case for Slot Attention. By only evaluating in the DINOSAUR setting, it is unclear how reliant the performance of the proposed model is on the pretrained ViT features.
2. In Figure 3 and 5.B, it seems the slots are sometimes reassigned before and after the self-modulation step? For example, the skater in Figure 3 is captured by the 4th slot before the modulation and the 5th slot after the modulation. The 5th slot before the modulation seems to capture part of the background, not any part of the skater. It is unclear to me why this would happen during the modulation update.

**Questions:**

1. Since the slots also contain position information (”where” information), it seems possible that this gets captured in the learned codebook. Is this something the authors observed?
2. How cherry-picked is Figure 2? Are there cases where these codes do not correspond to the same semantic concept?
3. In Figure 2, it seems that sometimes, multiple objects with similar semantics (e.g. multiple zebras or signs) are being captured by one slot. Did you notice your proposed method does more semantic grouping instead of instance grouping compared with vanilla DINOSAUR?

**Limitations:**

Yes

---

> ### Author Rebuttal · Authors · 2024-08-07
>
> ## Experiment on models other than DINOSAUR
> We appreciate your suggestion for additional experiments. We have implemented our self-modulation technique with the original slot attention setting [28], which includes training encoders from scratch and using an image reconstruction objective. The following table summarizes the results on the CLEVR6 dataset:
> |  | FG-ARI | mBO |
> |----------|----------|----------|
> | Original slot attention | 98.7 | 21.2 |
> | Original slot attention + SelfMod | 88.7 | 61.9 |
>
> We observe a significant improvement in mBO, showing that our self-modulation technique is applicable to slot attention and provides complementary benefits. Although FG-ARI decreased, mBO is considered more robust and should be prioritized when evaluating model performance, as detailed in our response to reviewer 3ckb ("Necessity & effect of VQ: (3) Difference in FG-ARI and mBO in VQ Ablation"). We have also included qualitative results in Figure R3 of the attached PDF, which demonstrates that self-modulation markedly improves segmentation. These results indicate our method's effectiveness with different encoder configurations and training objectives, suggesting it does not heavily rely on pretrained features. We will add the results and discussion in the revision.
> ## Slots reassigned to other object after modulation
> Thank you for your acute observation. We sincerely apologize for the discrepancy in Figures 3 of the main paper and B.5 of the appendix, which resulted from inadvertently pairing slot attention maps from different samples: the slot attention maps after self-modulation are correct but attention maps before self-modulation were wrongly paired. We have provided the updated figures as Figure R1 and R5 in the attached PDF.
>
> As the reviewer correctly pointed out, slots are modulated based on the information captured during the first slot attention. In the corrected Figure 3 (Figure R1 of the attached PDF), row 3, the 5th slot, which initially captures the skater with less certainty, is modulated to capture the skater with higher confidence by leveraging the semantic and spatial information from the initial slot.
>
> We would like to reaffirm that our original observation—“ modulation dynamically refines the attention maps, depending on how well they have captured the scene”—remains valid. For instance, in the 2nd row, the 1st slot initially captures both the plates and the orange, but after modulation, it is associated exclusively with the orange, yielding the plates to the 7th slot which more accurately captures the plates during the first slot attention. Similarly, in the 4th sample, the 3rd slot, which initially associates with the straw in a trivial manner, is modulated to identify and locate the straw more accurately. On the other hand, the attention maps of the 1st row are refined at the boundaries only since the initial attention maps were well structured.
>
> Once again, we apologize for any confusion caused by this error. We appreciate your understanding and the opportunity to clarify our findings.
>
> ## Does the codebook also contain positional information?
> While slot representations can indeed contain positional information due to the reconstruction objective, our observations suggest that the codebook primarily encodes semantic content. Performing $K$-means clustering on slot representations from DINOSAUR, we observed that the clusters are primarily grouped by semantic categories rather than spatial positions, as shown in Figure R6 of the attached PDF. This indicates that semantic information dominates in slot representations. Figure 2 in the main paper further supports this, showing objects with similar semantics but different positions mapped to the same code. Thus, while present, positional information likely occupies a small subspace in slot representations, with semantic content being the primary factor captured by our codebook.
>
> ## Question about codebook visualization/semantics
> Figure 2 is not heavily cherry-picked. Most codes consistently represent a single semantic concept, with a few codes that portray supercategories such as animals and humans (people and their hands), as shown in Figure R4 of the attached PDF.
>
> Also, we did notice an interesting edge case where some codes capture only the top-left-most patch, as shown in Figure R2 of the attached PDF. We attribute this to the autoregressive decoding process. During decoding, the top-left patch must be reconstructed first without any context from previous patches. Consequently, in images with less visual complexity, some slots appear dedicated to capturing these top-left patches. However, these special codes are minimal (1 or 2 out of 512 codes) and have limited impact. We will include more codebook visualizations and a detailed discussion in the revision.
>
> ## Do modulated slots perform semantic grouping more than instance grouping?
> Thank you for the insightful observation. To quantitatively assess whether our method favors semantic grouping over instance grouping compared to DINOSAUR, we conducted an analysis using two metrics:
> Instance Precision: The precision score with instance-level GTs
> Semantic Recall: The recall score with semantic segmentation GTs
> If our model was biased toward semantic grouping, we would expect to see a lower instance precision (as semantic segmentation ground truth masks often have larger sizes) and a higher semantic recall compared to DINOSAUR. However, our analysis on the COCO dataset shows that our method improves both metrics, as shown in the table below.
> |  | Instance Precision | Semantic Recall |
> |----------|----------|----------|
> | DINOSAUR | 41.1 | 75.0 |
> | Ours | 42.3 | 75.3 |
>
> These results suggest that the proposed top-down pathway enhances identifying both individual instances and semantic categories rather than favoring one.

---

> > ### Comment · Reviewer_Yith · 2024-08-12
> > **Reply to Rebuttal**
> >
> > Thank you for the rebuttal and running additional experiments. The updated figures for the slots before and after modulation make much more sense now. The additional results with vanilla Slot Attention on CLEVR are also promising, although I would encourage the authors to run additional seeds and/or make sure the hyperparameters are correct, as I've seen better results with vanilla slot attention on CLEVR (looking at the qualitative results it seems the background is being split). For what it's worth, I think it would still be a useful contribution if the proposed approach only improves the metrics on more complex datasets (since as a field, we should be generally pushing towards more complex datasets), but it is important to know the effect on simpler datasets as well so researchers are aware of the limitations. I would encourage the authors to include these results in the final version of the paper. After reading the rebuttal as well as the other reviews and rebuttals, I have decided to increase my score to 7.

---

> > > ### Author Response · Authors · 2024-08-13
> > >
> > > We appreciate your thorough review and time evaluating our rebuttal. We're pleased our updated figures and CLEVR experiments have resolved your concerns. We acknowledge your suggestion and will include CLEVR results from multiple seeds in the final paper. Thank you for your constructive feedback and recognizing the strength of our work.

---

### Official Review · Reviewer_3ckb · 2024-07-12

**Soundness:** 3
**Presentation:** 3
**Contribution:** 3
**Rating:** 6
**Confidence:** 4

**Summary:**

This work proposes a novel unsupervised object-centric learning method. In particular, the proposed approach enhances the slot-attention mechanism by incorporating a "top-down pathway" that highlights features relevant to objects in the image.

The method first employs a standard self-attention mechanism to extract slot vectors and the corresponding attention masks. These initial slot vectors and attention masks are then fed into another round of the slot-attention mechanism. In this round, they modulate the value maps of the cross-attention mechanism separately for each slot. This modulation is channel-wise for slot vectors and spatial-wise for attention masks, enabling the slot-attention mechanism to emphasize to features relevant to the slots discovered on the first slot-attention round.

Furthermore, the method quantizes the slot vectors from the first slot-attention round using the VQ approach before utilizing them for modulation in the second round.

To evaluate the proposed self-modulated-based slot-attention mechanism, the authors integrate it into the DINOSAUR framework and apply it to the COCO, VOC, MOVIE-E, and MOVIE-C datasets for the downstream task of object segmentation. The results demonstrate improvements over the vanilla slot-attention mechanism. Ablation studies are conducted on the COCO dataset.

**Strengths:**

STRENGTHS:
- The idea of modulating the value maps for each slot, using slots found from a previous self-attention round, is interesting. This technique allows the subsequent self-attention round to focus more on features that are relevant to the objects discovered in the initial round. Essentially, it's an additional iterative improvement for the extracted slots, supplementing the existing iterative mechanism in the standard slot-attention.
- The proposed method has been demonstrated to improve object segmentation results on all tested datasets. Furthermore, the authors utilized challenging datasets, such as COCO, to validate the effectiveness of their method, rather than resorting to simpler one, which is often among other object-centric works.
- The paper has strong results and detailed ablation studies.

**Weaknesses:**

- **(W1)** It's unusual that the mIoU results in the MOVIE-C and MOVIE-E datasets surpass the mBO results.  The mBO is computed by assigning each ground truth mask the predicted mask with the largest overlap, and then averaging the IoUs of the assigned mask pairs. On the other hand, the mIoU metric is more strict and employs Hungarian matching (rather than greedy matching) to assign predicted masks to the ground truth masks. Consequently, mBO results should be either greater than or equal to the mIoU results. Therefore, it's highly likely that there's a bug in the computation of the mBO or the mIoU metrics on the MOVIE-C and MOVIE-E datasets. Or a typo / mistake when copying the results to the paper maybe.
- As a relevant side note for the upcoming comments, it's important to mention that the FG-ARI metric is generally viewed as unreliable for assessing unsupervised object-centric methods. This is because it only takes foreground pixels into account, which can provide a deceptive understanding of segmentation quality by disregarding the localization accuracy of predicted masks [A, B, C, 30, 42].
-  **(W2)** The motivation behind the Vector Quantization (VQ) component of the method is unclear. In particular it's uncertain why quantizing the slot vectors is essential for self-modulation. In fact, Table 4 indicates that the method is sensitive to the codebook size (also acknowledged by the authors in the limitations section). The mBO performance for sub-optimal codebook sizes 128, 256, and 1024 (30.2%, 30.5%, and 29.8%, respectively) is either the same or worse than the reproduced DINOSAUR mBO performance (30.5% from Table 6). Conversely, removing VQ achieves 32.3%, which is only marginally worse than the 32.7% obtained with VQ and the optimal codebook size. Therefore, VQ unnecessarily complicates the proposed method. As mentioned earlier, FG-ARI is unreliable, and the higher differences in this metric (36.3% w/o VQ vs 37.7% w/ VQ) should be disregarded.
- It's also somewhat concerning that Table 6 shows only a minimal impact on mBO performance when using channel-wise modulation (from 32.5% to 32.7%). However, this does not really increase the complexity of the method, as VQ does.
- **(W3)** Lacks discussions on how the proposed method affects the training and test time
- **(W4)** Table 3 is missing comparisons with related works, such as Rotating Features [D] and SPOT [C], which also demonstrate strong results.

[A] Genesis: Generative scene inference and sampling with object-centric latent representations, ICLR 2020.
[B] Unsupervised Layered Image Decomposition into Object Prototypes, ICCV 2021.
[C] SPOT: Self-Training with Patch-Order Permutation for Object-Centric Learning with Autoregressive Transformers, CVPR 2024
[D] Rotating features for object discovery. NeurIPs 2023.

[30] Bridging the gap to real-world object-centric learning, ICLR 2023.
[42] Slotdiffusion: Object-centric generative modeling with diffusion models, NeurIPs 2023.

**Questions:**

- My primary concern regarding this work (mentioned as (W2) in the weaknesses section) is that the channel-wise and Vector Quantization (VQ) components unnecessarily complicate the method without demonstrating significant improvements in the ablation results. I would appreciate it if the authors could provide compelling arguments, such as additional experimental evidence on other datasets (e.g., VOC or MOVIE-E/C) or different settings (e.g., longer training), that show (if it is the case) the benefits of channel-wise modulation with VQ.

- Please address the concerns (W1), (W3), and (W4) mentioned in the weaknesses section.

**Limitations:**

Yes.

---

> ### Author Rebuttal · Authors · 2024-08-07
>
> ## Validity of mIoU scores in MOVI-C/E
> Thank you for your astute feedback. We sincerely apologize for the mistake in the mIoU calculations on the MOVI datasets and any confusion it may have caused. As you correctly pointed out, the mBO should be greater than or equal to the mIoU. The reported mIoU is inflated due to omitting the first ground-truth mask. We have provided the corrected mIoUs in Table R1 of the attached PDF. Importantly, despite this error, our revised results still demonstrate that our method outperforms DINOSAUR, substantiating our original claim about the benefits of our top-down pathway. We greatly appreciate your diligence in identifying this issue and allowing us to correct it.
>
> ## Necessity & effect of VQ
> Thank you for your insightful comment regarding Vector Quantization (VQ). We explain the necessity of VQ by addressing the inquiries below:
>
> **(1) Necessity of VQ for Self-Modulation**
>
> VQ is fundamental to our method since it allows the model to capture recurring semantic patterns from rough object representations produced by the first slot attention. Similar to online clustering, VQ maps a continuous slot representation to the nearest discrete code. Throughout this process, codes in the codebook are updated to encode only the semantic information from the noisy object representations; this is consistent with clustering being used to capture high-level semantic knowledge from noisy inputs in various tasks [D, 44]. Therefore, incorporating VQ into the top-down pathway enables our model to focus on core semantic concepts of the dataset, which are bootstrapped from noisy representations without extra supervision. This leads to more refined object-centric representations.
>
> **(2) Sensitivity to Codebook Size**
>
> We acknowledge the performance dependence on the codebook size, as noted in our limitations section. This characteristic is indeed common in clustering-based methods.
> However, we would like to emphasize that the optimal codebook size was determined automatically using the training set only (without the validation set). To choose the codebook size, we monitor the perplexity of the code usage distribution during training. Perplexity—the exponent of entropy—indicates how uniformly the codes are being used and typically increases as the codebook becomes larger. However, when the codebook size exceeds the number of semantic patterns in the data, some codes become neglected, causing low code utilization [E, 45, 26]. Hence, the ideal codebook size leads to the maximum perplexity.
> To find the optimal codebook size, we start with a size of 64 and monitor the perplexity of the codebook over training. After sufficient iteration (e.g., 250K), the codebook size is doubled repeatedly until the perplexity reaches a plateau. For example, on COCO, the perplexity when the codebook size is 256, 512, and 1024 are 176.9, 253.9, and 242.8, respectively, where 512 was chosen as the final size. Thereby, an adequate codebook size could be efficiently determined without using the validation set.
>
> **(3) Difference in FG-ARI and mBO in VQ Ablation**
>
> While we agree that mBO is the more reliable metric, FG-ARI can provide complementary information when mBOs are similar. FG-ARI is considered deceptive because it does not penalize under-segmentations that include background pixels [A], sometimes resulting in misguidedly high FG-ARIs despite low mBOs. However, in Table 6 of the main paper, VQ improves both metrics. The increased mBO, which accounts for background pixels, suggests that the FG-ARI increase is not due to under-segmentation but due to improved foreground object segmentation. Thus, we respectfully claim that the increase in FG-ARI with VQ should be considered as valid evidence for the effectiveness of VQ, especially considering the slight improvement in mBO.
>
> **(4) VQ Ablation with Full Training Schedule**
>
> We further validate the effectiveness of VQ with an experiment under a full training schedule (500K iterations) on COCO as suggested. The results demonstrate consistent improvement in both FG-ARI and mBO, clearly showing that VQ plays a crucial role in our model:
> |  | FG-ARI | mBO$_i$ |
> |----------|----------|----------|
> | Ours wo/ VQ | 36.3 $\pm$ 0.5 | 32.4 $\pm$ 0.1 |
> | Ours | 37.4 $\pm$ 0.0 | 33.0 $\pm$ 0.3 |
>
> In summary, while we acknowledge the sensitivity to codebook size, we believe the benefits of VQ outweigh this limitation. Additionally, an appropriate codebook size can be efficiently determined by monitoring perplexity during training.
>
> ## Computation cost analysis
> While our model requires one more forward pass for slot attention, the additional computation cost is negligible. Slot attention accounts for only 0.64% of the total FLOPs, compared to 71.26% for the encoder and 28.10% for the decoder. Thus, our model requires 47.62 GFLOPs while DINOSAUR needs 47.32 GFLOPs. Also, processing the entire COCO 2017 val split takes 71.4 seconds for our model, while 70.5 seconds in the DINOSAUR.
> ## Missing comparison with recent work
>
> Thank you for pointing out the missing related work. We will add comparisons with Rotating Features [C] and SPOT [B] in the revision. We would like to kindly remind the reviewer that SPOT was published at CVPR 2024, after our submission deadline. Moreover, SPOT's self-training and sequence permutation techniques are completely orthogonal to our work on top-down pathways, suggesting potential for future integration of these complementary approaches.
>
> ---
> [A] Genesis: Generative Scene Inference and Sampling with Object-centric Latent Representations, ICLR, 2020.
>
> [B] SPOT: Self-Training with Patch-Order Permutation for Object-Centric Learning with Autoregressive Transformers, CVPR, 2024
>
> [C] Rotating Features for Object Discovery, NeurIPS, 2023
>
> [D] Deep Clustering for Unsupervised Learning of Visual Features, ECCV, 2018
>
> [E] Straightening Out the Straight-Through Estimator: Overcoming Optimization Challenges in Vector Quantized Networks, ICML, 2023

---

> > ### Comment · Reviewer_3ckb · 2024-08-13
> >
> > I thank the authors for the detailed responses to my comments. Their response  addresses my main concerns regarding the necessity of VQ: the codebook size selection is automatic and uses the training split, and VQ offers bigger improvement for longer training. Therefore, and after reading the other reviews and the respective rebuttals, I am going to increase my score.

---

> > > ### Author Response · Authors · 2024-08-14
> > >
> > > We sincerely appreciate your careful review of our responses and your reconsideration. We're glad our explanations about codebook size selection and VQ benefits clarified your concerns. In our future revision, we will include discussions and experiments about VQ, which will significantly improve the paper's quality. Thank you for your valuable feedback!

---

### Author Rebuttal · Authors · 2024-08-07

We sincerely thank all the reviewers for the constructive and insightful comments. Our work introduces a novel top-down pathway for object-centric learning that consistently improves performance across multiple benchmarks. Reviewers highlighted several strengths of our work, including the clarity of the manuscript, proposed method’s motivation and novelty, consistent improvements over the baseline across challenging real-world datasets, and extensive analyses.

In each response to the reviewer, we have carefully addressed every comment and question to the best of our ability, providing additional results, clarifications, and analyses where requested. Key points we elaborate on include the clarification for vector quantization, comparisons to additional baselines, and further insights into our method's behavior. We also provide additional results and visualizations in the attached PDF.

Due to the word limit, we referenced **papers cited in the main paper with numbers** and **newly cited ones with alphabetical letters**.

Thank you again, and we look forward to a constructive interaction in the following discussion period!

---

### Decision · Program_Chairs · 2024-09-25

**Decision:**

Accept (poster)

**Comment:**

Slot attention has emerged as a popular approach for unsupervised object discovery, and the improvements presented in this paper have been appreciated by all the reviewers. Given that the work is both timely and well-executed, I recommend accepting it.

The key highlights include the use of a visual codebook to self-modulate internal activations by defining the "what" and "where" information based on the attention maps. As noted by reviewer qEia, the model consistently outperforms the underlying DINOSAUR model on which this technique is built. Both the qualitative and quantitative evaluations are impressive. Reviewer YiTH and PJTV has offered valuable suggestions on how the paper could be enhanced by incorporating insights from related work. I hope the authors take these into consideration while preparing the final version.